# Rhodium nanocrystals on porous graphdiyne for electrocatalytic hydrogen evolution from saline water

Yang Gao[1], Yurui Xue [1,2] ✉, Lu Qi[2], Chengyu Xing[1], Xuchen Zheng[1,3], Feng He [1] ✉ & Yuliang Li [1,3] ✉

The realization of the efficient hydrogen conversion with large current densities at low overpotentials represents the development trend of this field. Here we report the atomic active sites tailoring through a facile synthetic method to yield well-defined Rhodium nanocrystals in aqueous solution using formic acid as the reducing agent and graphdiyne as the stabilizing support. High-resolution high-angle annular dark-field scanning-transmission electron microscopy images show the high-density atomic steps on the faces of hexahedral Rh nanocrystals. Experimental results reveal the formation of stable $sp$−C-Rh bonds can stabilize Rh nanocrystals and further improve charge transfer ability in the system. Experimental and density functional theory calculation results solidly demonstrate the exposed high active stepped surfaces and various metal atomic sites affect the electronic structure of the catalyst to reduce the overpotential resulting in the large-current hydrogen production from saline water. This exciting result demonstrates unmatched electrocatalytic performance and highly stable saline water electrolysis.

Directly producing green hydrogen ($H_2$) on large-scale from seawater electrolysis is a long-sought dream of mankind[1–7]. Accompanied by technological advances that make it possible to produce $H_2$ on a large scale from electrolysis of water, however, the challenge is the lack of a catalyst to realize efficient seawater splitting for large-scale $H_2$ production with large current densities at low overpotentials that can meet the industrial requirements, without deactivating the catalysts by the ions of sodium, chlorine, calcium, chloride, and other components in seawater[8–12]. To overcome these scientific challenges, a variety of highly efficient catalysts have been designed and synthesized by researchers (e.g., nitrides[9], transition metal hydroxides[10], and phosphide[11,12], etc.). Although previously reported catalysts demonstrated high performance under some conditions, they were far from our expectations in terms of overall performance. Most catalysts exhibit large overpotential and low long-term stability at large current densities which is difficult to meet the industrial demand. The

development of intrinsically active catalysts capable of large-scale hydrogen production at low overpotential is a key step to achieve zero pollution and hydrogen fuel highly economy. These serious issues force us to find a way in the design of catalysts to explore the highly matched structure, band gap, and intrinsic activity of the composite catalyst for application on large scale of hydrogen production at low overpotential.

Given an accurate understanding of the source of its intrinsic catalytic activity, a catalyst with simple composition is desirable to help understand its mechanism. After in-depth consideration and analysis, we propose that such an ideal catalyst should have larger current density at lower overpotentials, resistance to seawater induced inactivation, and excellent long-term stability. From the further perspective of mass hydrogen production through water splitting, the electrocatalysts should have the following necessary properties: (i) high intrinsic catalytic activity that can achieve large current density

[1]CAS Key Laboratory of Organic Solids, Institute of Chemistry, Chinese Academy of Sciences, Beijing, PR China. [2]Science Center for Material Creation and Energy Conversion, Institute of Frontier and Interdisciplinary Science, School of Chemistry and Chemical Engineering, Shandong University, Jinan, PR China. [3]University of Chinese Academy of Sciences, Beijing, PR China. ✉e-mail: yrxue@sdu.edu.cn; hefeng2018@iccas.ac.cn; ylli@iccas.ac.cn

HER at low overpotentials under saline water conditions; (ii) strong corrosion resistance and electrochemical/mechanical stability at large current densities; (iii) efficient gas/mass diffusion ability capable of separating the formed H$_2$ bubbles to maintain the catalytic activity for large current density HER; (iv) long-term operational stability under practical conditions (e.g., saline water, strong basic water, or sea-water); and (v) low cost.

To sum up the previous experience, the catalytic activity material should be neither too strong nor too weak in combination with the reaction intermediates. Rhodium (Rh) is the nearest metal to Platinum (Pt) located at the vertex of volcano plot with small Gibbs free energy[13], and has been demonstrated have excellent catalytic stability. Distinct from bulk metals, the intrinsic activity of the catalyst has been demonstrated to strongly depend on the size, the exposed defects, and the supports[14–19]. As an important traditional metal support, carbon materials show great application value in catalysis field, therefore, carbon materials are considered as the key component of high-performance catalysts[20–22]. The shortcoming of the catalysts composed of traditional carbon materials, however, generally result in ambiguous chemical and electronic structures due to the hash and complicated pyrolysis process, the accurate understanding of the catalytic mechanisms is limited which leads to the inability to optimize the catalyst efficiently and accurately to obtain high-performance catalyst.

Comprehensive analysis of the current carbon materials, porous graphdiyne (GDY) shows the advantages, 2D layered structure with *sp*- and *sp$^2$*-hybridized carbon atoms, large porous structure, infinite distribution of active sites, excellent electron transfer ability, high surface area and high stability[23–43]. These natural advantages make GDY as a catalyst with excellent comprehensive performance for hydrogen evolution reaction (HER)[30–35], oxygen reduction reaction (ORR)[36,42], oxygen evolution reaction (OER)[34,35], overall water splitting (OWS)[34,35], nitrogen reduction reaction (NRR)[37,38], CO$_2$ reduction reaction (CO$_2$RR)[39], methanol oxidation reaction (MOR)[40] and CO catalytic oxidation[41]. In fact, the previous reports have demonstrated GDY to be

an ideal catalyst support for selectively anchoring single metal atoms and controllably regulating the coordination environments of metal atoms toward highly catalytic activity. A variety of porous GDY-based catalysts have been developed, e.g., GDY-based zero-valent atomic catalysts[30–32], heterostructured catalysts[34,37], quantumn dots catalysts[38,40–42], and metal-free catalysts[35,36].

Here, we show a concise route to realize the controllable synthesis of well-defined Rh nanocrystals with high-density of atomic defects through a facile method in aqueous solution using the formic acid as reduction agent and GDY as the stabilizing support (Fig. 1a). HAADF-STEM images clearly show the presence of the atomic steps. Experimental and theoretical results demonstrate that the stepped surface and special interactions form between GDY and Rh endow the catalyst with excellent electrocatalytic activity and stability for large current density H$_2$ production from alkaline simulated seawater (saline water). For example, the Rh/GDY can deliver 1000 mA cm$^{-2}$ at a very small overpotential of 65 mV vs. RHE, which is better than the reported electrocatalysts, and commercial Pt/C electrocatalyst.

## Results

### Electrocatalysts synthesis and characterization

Prior to the controlled synthesis of well-defined porous Rh nanocrystals, a three-dimensional (3D) flexible porous GDY electrode was synthesized by growing a film of GDY nanosheets array on the surface of 3D carbon fiber cloth (CC), at ambient temperatures and using hexaethynylbenzene (HEB) as the precursor. Scanning electron microscopy (SEM, Supplementary Fig. 1) and high-resolution transmission electron microscopy (HRTEM, Supplementary Fig. 2) images reveal that the ultrathin GDY nanosheets are interconnected with each other and self-supported on the CC substrates, resulting a porous morphology with enlarged surface area and more exposed active sites which benefits the catalytic performance. The 3D porous GDY was immersed into an aqueous solution containing RhCl$_3$ and HCOOH for the in-situ adsorption of Rh atoms as the starting sites for subsequent nucleation and growth of the Rh nanocrystals on the surface of GDY

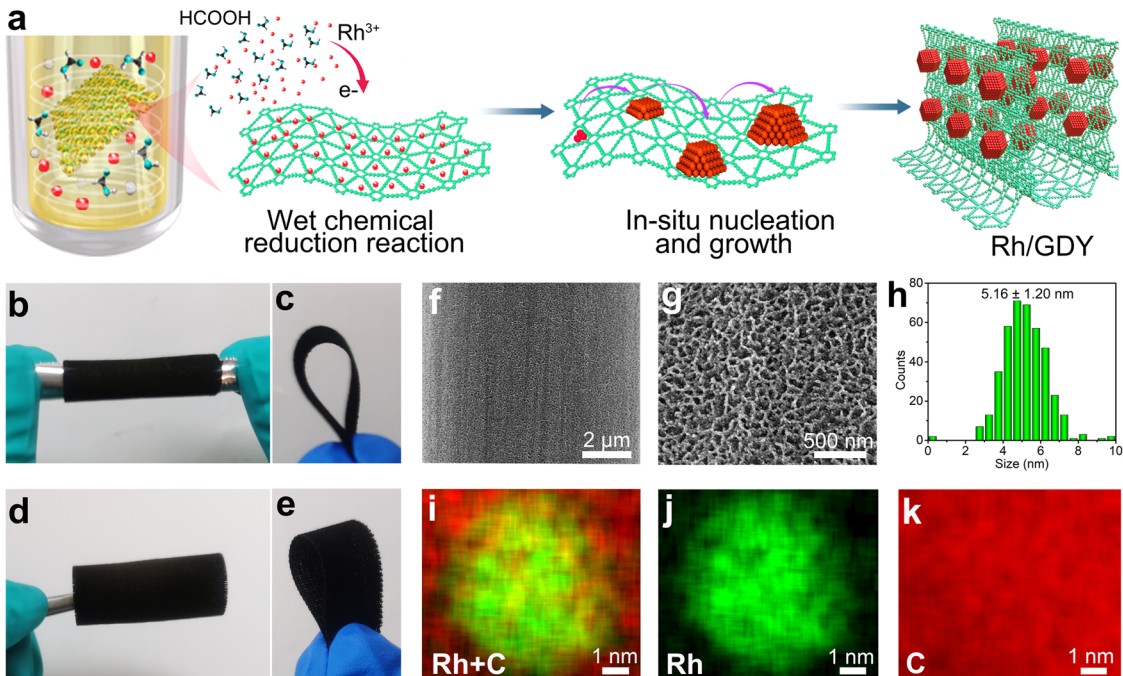

**Fig. 1 | Synthesis and morphological characterization of Rh/GDY. a** Synthesis routes to porous Rh/GDY electrodes. Optical images of **b**–**e** Rh/GDY electrodes with various bending morphologies. **f** Low- and **g** high-magnification SEM images of Rh/GDY. **h** Size distribution of Rh nanocrystals on the surface of GDY (403 Rh nanocrystals were counted). **i-k** EDS elemental mapping images of **i** overlapping, **j** Rh and **k** C elements on Rh/GDY.

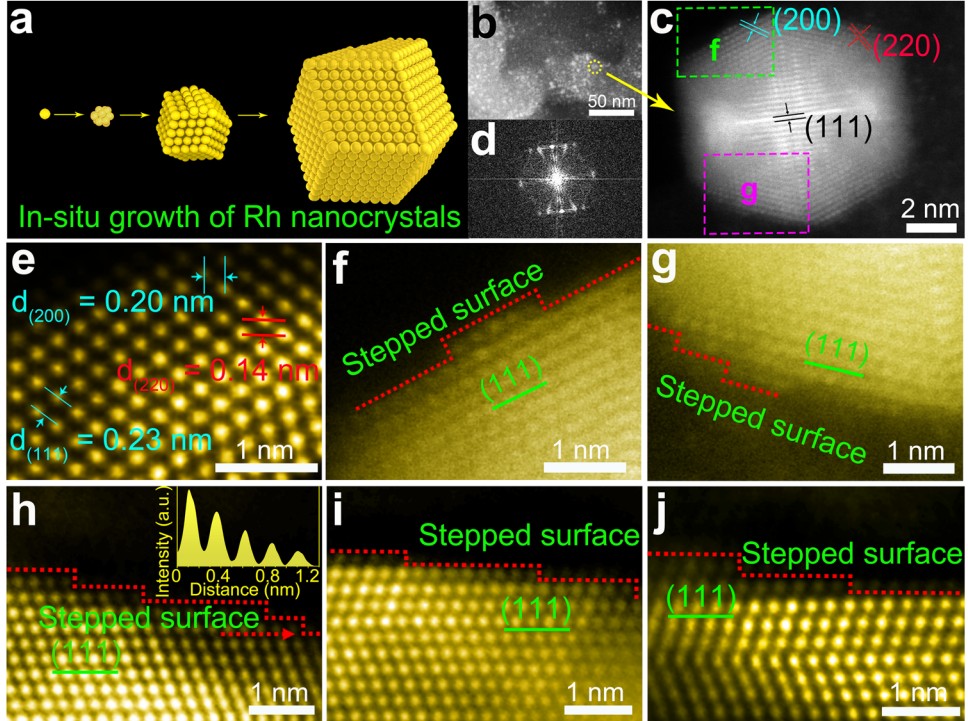

**Fig. 2 | Atomic scale HAADF-STEM characterization of Rh/GDY. a** Simulated atomic arranging process and model of individual Rh nanocrystal. **b** HAADF-STEM image of Rh/GDY. **c** High-resolution HAADF-STEM image of a single Rh nanocrystal grown on GDY. The labelled squares indicate the areas magnified in the following observations. **d** Fast Fourier transform pattern of Rh/GDY from **c. e** HAADF-STEM image of Rh/GDY. **f** HAADF-STEM image of a (111) plane from area f marked in **c.** Terraced atomic layers represent a stepped surface. **g** HAADF-STEM image of a (111) plane with atomic kinks from area g marked in **c. h–j** HAADF-STEM images of the (111) plane, the intensity profile (inset of **h**) along the dotted line represents a stepped surface along the edge.

(Rh/GDY; Fig. 1a). Highly flexible Rh/GDY electrodes (Fig. 1b–e) with 3D porosity (Fig. 1f, g) were obtained. Note that these features make them more suitable to meet the requirements of practical uses for assembling electrolysis devices. HAADF-STEM (Supplementary Fig. 3) and HRTEM images (Supplementary Fig. 4a) confirmed the homogeneous dispersion of Rh nanocrystals on the surface of GDY nanosheets with a narrow size distribution of $5.16 \pm 1.20$ nm (Fig. 1h). STEM (Supplementary Fig. 5) and corresponding energy dispersive X-ray spectroscopy (EDS) mapping images (Fig. 1i–k) show the uniform dispersion of Rh and C elements. Inductive coupled plasma mass spectrometry (ICP-MS) result shows the mass loading of Rh on GDY is 0.244 wt% (Supplementary Table S1). Raman spectra of the Rh/GDY system (Supplementary Fig. 6) unambiguously show the characteristic peaks of conjugated diyne links (1964 and 2170 $cm^{-1}$), indicating that the GDY structure was retained during the synthesis process. Compared with pure GDY (Supplementary Fig. 6), shifts in D band (1365 $cm^{-1}$) and G band (1585 $cm^{-1}$) suggest the existence of the interactions between Rh and GDY. The intensity ratio of the D/G bands in the Raman scattering is commonly used to assess the defects density present on carbon materials. The larger D/G ratio for the Rh/GDY system (0.63) than pure GDY (0.55) indicates the higher defect density of Rh/GDY. X-ray diffraction (XRD) patterns in Supplementary Fig. 7 shows typical diffraction patterns of Rh/GDY at approximately 41°, 48°, 70°, and 84° corresponding to the metallic Rh (JCPDS no. 05-0685), and two broad diffraction peaks centered at 20–30° and 44° could be assigned to GDY species.

High-resolution HAADF-STEM and TEM images were used to reveal the atomic scale structure of Rh/GDY. The arrangement of Rh atoms on the surface of GDY resulted in a polyhedral morphology (Fig. 2a, b; HAADF-STEM images from two different sites are shown in Fig. 2f, g) with fringe distance of 0.23, 0.20 and 0.14 nm corresponding to the (111), (200) and (220) planes of Rh, respectively (Fig. 2c, e,

Supplementary Fig. 4b, c). Fast Fourier transform (FFT) pattern (Fig. 2d) reveals that the equilibrium shapes might be the truncated octahedrons (Supplementary Fig. 8). HAADF-STEM images reveal that the edges of (111) planes are not straight but contain numerous of mono-layered (Fig. 2f) and multi-layered steps (Fig. 2g–j). HAADF results reflect that the GDY has high affinity with Rh atoms which facilitates the selective and stable anchoring of Rh atoms. Besides, the high intrinsic reductivity of GDY controllably regulates the interface of Rh growth, leading to the controllable stepped morphology of the Rh nanostructures. Intensity changes shown in the inset of Fig. 2h indicates the presence of large amounts of atomic steps. In addition, HAADF-STEM images show that the high density of atomic steps contains different atoms at the "corner" and "surface" sites, respectively. The difference between the "corner" and "surface" sites in stepped Rh/GDY can be clearly identified by the coordination number of Rh atoms (7-coordinated corner atoms and 9-coordianted surface atoms, Supplementary Fig. 9). Note that low-coordination metal atoms have demonstrated can interact more strongly with reactants due to the modified local electronic structure[42], which helps to reduce the reaction barriers compared to the atoms at surface sites. The atoms at the corner sites possess low-coordination are therefore proposed to be the main origins of the high HER catalytic activity of Rh/GDY.

Depth-profiling X-ray photoelectron spectroscopy (XPS) coupled with ion sputtering, which is important and necessary for identifying the active sites, was used for the high-resolution analysis of the atomic composition and chemical states of Rh/GDY from the surface to inner part (Fig. 3a). The Rh 3d obtained from different samples (Fig. 3b, c) and C 1 s (Fig. 3d; Supplementary Fig. 10) spectra were analyzed and showed that there are no changes in the compositions and chemical states in the Rh/GDY from the surface to inner part. The slight changes in the shape of the XPS peaks for sample 1 could be due to the rough surface structure of the catalysts or the surface charging effects during

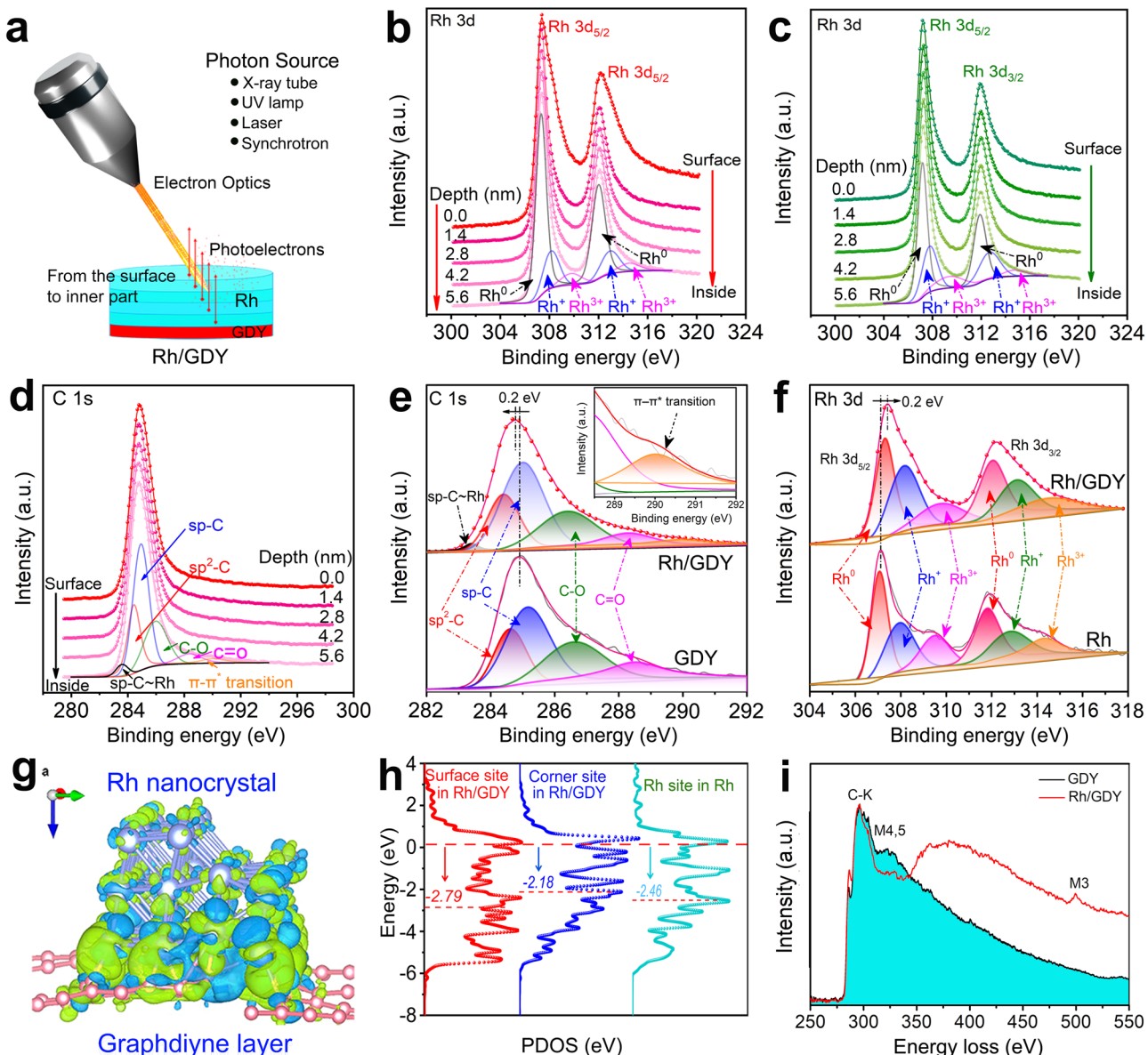

**Fig. 3 | XPS and PDOS characterizations of the samples. a** Schematic representation of the depth profiling experiments. The sets of **b, c** Rh 3d obtained from different samples and **d** C 1 s XPS spectra of Rh/GDY measured during the depth profiling experiments. The high-resolution **e**, C 1 s, **f** Rh 3d XPS spectra of the catalysts. **g** The differential charge density of Rh/GDY. Image created with VESTA software. **h** Projected density of state (PDOS) of Rh/GDY and Rh. **i** Electron energy-loss spectroscopy (EELS) of carbon and rhodium elements in the GDY and Rh/GDY.

the depth-profiling XPS tests. The high-resolution XPS analysis shows that C 1 s XPS spectra (Fig. 3e) show six peaks at 283.4 ($sp$−C-Rh), 284.4 ($sp^2$-C), 285.0 ($sp$-C), 286.4 (C−O), 288.3 (C=O), and 290.0 eV (π−π* transition) for Rh/GDY, while only four peaks at 284.6 ($sp^2$-C), 285.2 ($sp$-C), 286.6 (C−O), 288.5 eV (C=O) can be observed for pure GDY[26]. The formed $sp$−C-Rh and π−π* transition peaks indicate the formation of chemical interactions between Rh and GDY[26,37,44]. Our experimental results show that the formation of stable $sp$−C-Rh bonds can stabilize Rh nanocrystals and further improve charge transfer ability in the system.

As shown Fig. 3f, the high-resolution Rh 3d XPS spectra of Rh/GDY can be deconvoluted into three doublets peaks at 307.3, 308.2, and 309.7 eV for Rh $3d_{5/2}$ and 312.0, 313.0, and 314.5 eV for Rh $3d_{3/2}$, corresponding to the $Rh^0$ (307.3/312.0 eV), $Rh^+$ (308.2/313), and $Rh^{3+}$ (309.7/314.5 eV), respectively[15]. In our experiments, GDY was used as the support for the in-situ growth of the Rh nanocrystals, including the adsorption of Rh ion on the surface of GDY and the following in-situ

nucleation and growth of Rh nanocrystals (Fig. 1). Due to the reducibility of GDY itself and HCOOH, the Rh ions can be easily reduced to metallic species. Besides, the presence of obvious charge transfer from Rh to GDY make partial Rh species with higher valence states. This result indicates that we obtained the Rh/GDY catalysts with mixed-valence states[25,37,40,45]. The spontaneous recycling of the $Rh^0/Rh^+/Rh^{3+}$, and the specific incomplete charge transfer between multiple valences metal species in this system can endow Rh/GDY with excellent catalytic activity.

In addition, the negative (0.2 eV) and positive (0.2 eV) shifts in the binding energies of C (Fig. 3e) and Rh (Fig. 3f) species demonstrate the efficient charge transfers from Rh to GDY, in consistent with the differential charge density distribution map results (Fig. 3g). We further compare the projected density of state (PDOS) of different sites in Rh/GDY and pure Rh metal, respectively (Fig. 3h). It is shown that the Rh $d$-center of surface site (i.e., −2.79 eV) and corner site (i.e., −2.18 eV) in Rh/GDY is lower and higher compared to that of Rh site in pure Rh

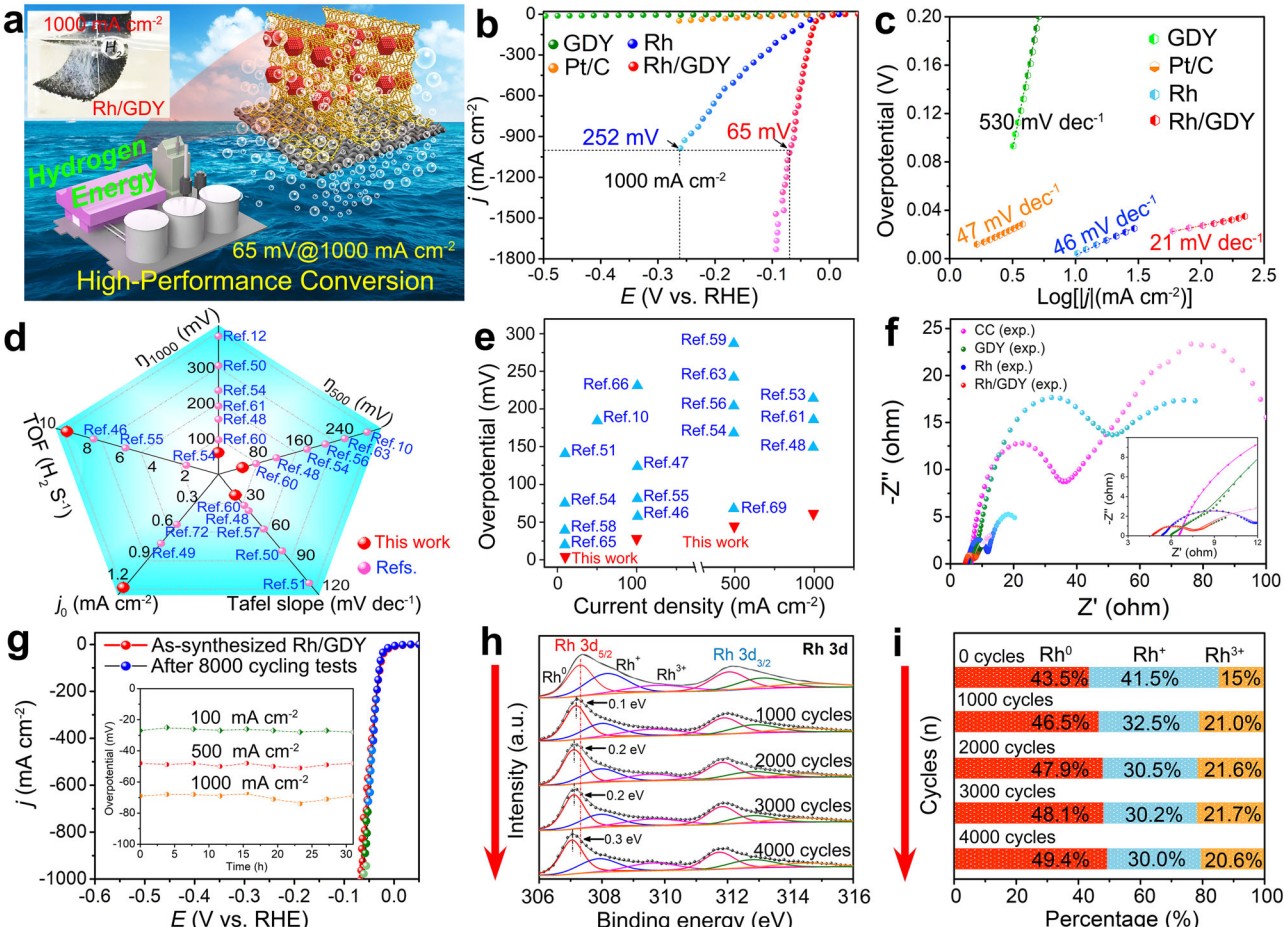

**Fig. 4 | HER performances of Rh/GDY in saline water. a** Schematic representation of efficient hydrogen fuel production on Rh/GDY. Inset: photograph of the hydrogen evolution on Rh/GDY electrode at 1000 mA cm$^{-2}$. **b** HER polarization curves and **c** corresponding Tafel plots of the as-synthesized catalysts. **d**, **e** Comparison of the HER catalytic activity of the Rh/GDY with the reported catalysts. **f** Nyquist plots of the catalysts. **g** Durability tests of Rh/GDY over 31 h at large current densities of 100, 500 and 1000 mA cm$^{-2}$, respectively. **h** Rh 3d XPS spectra of the Rh/GDY obtained after stability tests. **i** The atomic percentage of Rh$^0$, Rh$^+$ and Rh$^{3+}$ species during the stability tests.

metal (i.e., −2.46 eV), respectively. As known, the closer of *d*-center to Fermi level usually corresponds to the stronger adsorption ability of metal sites. Thereby, the corner site has much stronger adsorption ability than the surface sites. These show that GDY can efficiently regulate the electronic structures of stepped Rh metal that endow it excellent electrocatalytic activity and stability, verifying the vital role of GDY.

The electron energy loss spectroscopy (EELS) mapping of Rh nanocrystals grown on the GDY support (Fig. 3i, Supplementary Fig. 11) showed only signals for C and Rh, also indicating interaction between Rh and C in the Rh/GDY. Our results indicate the well-defined stepped surface characteristics and the strong interactions between Rh nanocrystals and GDY can result in obviously high charge transfer, which is beneficial to enhance the intrinsic HER activity of the catalyst.

**Electrocatalytic performances for HER in saline water**
Subsequently, the catalytic performance of Rh/GDY for H$_2$ fuel production from simulated seawater (i.e., H$_2$-saturated 1.0 M KOH + 0.5 M NaCl saline-alkaline electrolyte) was studied. All linear sweep voltammetry (LSV) curves were corrected with 100% *iR*-compensation. As observed from Fig. 4a and Supplementary Movie 1, Rh/GDY violently releases hydrogen bubbles and forms hydrogen clouds at a very small applied overpotential of 65 mV. The *iR*-corrected linear sweep voltammetry (LSV) curves of the samples indicated that Rh/GDY displays the best electrocatalytic activity with the smallest overpotentials (*η*) of

28, 48 and 65 mV at the current density (*j*) of 100, 500 and 1000 mA cm$^{-2}$, respectively (Fig. 4b), which are better than commercial 20 wt% Pt/C, Rh, GDY, CC, and reported electrocatalysts[4,44,46–66] such as Ni-SA/NC[4], Ru-Mo$_2$C@CNT[44], and Ru$_1$/D-NiFe[46] (Fig. 4d, e, Supplementary Fig. 12, Supplementary Tables 2 and 3). Rh/GDY in pure alkaline solution shows a similar HER catalytic activity to that obtained in the alkaline simulated seawater conditions (Supplementary Fig. 13). Besides, the as-prepared Rh/GDY catalyst exhibits better catalytic activity than those supported on other substrates such as graphene oxide, multi-walled carbon nanotubes, and glassy carbon electrode (Supplementary Figs. 14 and 15). All these findings confirmed the important role of GDY in enhancing the catalytic activity. Rh/GDY has the smallest Tafel slope of 21 mV dec$^{-1}$ for Rh/GDY (Fig. 4c), suggesting a underlaying Volmer-Tafel mechanism with discharging of two adsorbed hydrogen atoms (Tafel) as the rate-determining step (RDS)[67]. As the current density increases, mass transfer plays a key role in determining the current. We calculated the ratios of overpotential to current density (Δ*η*/Δlog|*j*|, R$_{η/j}$) for Rh/GDY in different current density ranges to evaluate the performance of a catalyst at large current densities more precisely. The ratio for Rh/GDY remains smaller (< 80 mV dec$^{-1}$; Supplementary Fig. 16) with the increase of the current density[53,68].

The exchange current density (*j*$_0$), mass activity (normalized to the Rh loading, 0.244 wt%), and turnover frequency (TOF) were further determined to reveal the intrinsic catalytic activities of the samples. As

shown in Fig. 4d, Rh/GDY exhibits a $j_0$ of 1.3 mA cm$^{-2}$, which is higher than that of the reported HER electrocatalysts in alkaline conditions such as RhPd-H/C (0.65 mA cm$^{-2}$)[14], $W_1Mo_1$-NG (0.26 mA cm$^{-2}$)[69], and ES-WC/$W_2$C (0.58 mA cm$^{-2}$)[70] (Supplementary Fig. 17; Supplementary Table 4). The TOF for Rh/GDY at an overpotential of 100 mV is 9.33 s$^{-1}$, remarkably larger than that corresponding to reported HER electrocatalysts, (e.g., $Ru_1$/D-NiFe LDH[46]: 7.66 s$^{-1}$; $Pt_{SA}$-NiO/Ni[55]: 5.71 s$^{-1}$; and $W_1Mo_1$-NG[69]: 0.42 s$^{-1}$; Fig. 4d; Supplementary Fig. 18; Supplementary Table 5). In addition, the Rh/GDY (274.6 A mg$_{Rh}^{-1}$) exhibits a higher mass activity than that of Rh (62.3 A mg$_{Rh}^{-1}$) at an overpotential of 50 mV (Supplementary Fig. 19; Supplementary Table 6).

The Nyquist plots were measured and fitted to a R(QR)(QR) equivalent circuit containing resolution resistance ($R_s$) and charge transfer resistance ($R_{ct}$). Rh/GDY showed the smaller $R_s$ (4.71 Ω) and $R_{ct}$ (2.84 Ω) than Rh ($R_s$ = 4.93 Ω, $R_{ct}$ = 9.81 Ω), GDY ($R_s$ = 5.70 Ω, $R_{ct}$ = 22.2 Ω) and CC ($R_s$ = 6.51 Ω, $R_{ct}$ = 89.6 Ω) (Fig. 4f, Supplementary Fig. 20; Supplementary Table 7), thereby showing high charge transport ability. The electrochemical active surface area (ECSA) of the catalysts was also evaluated from the double-layer capacitance ($C_{dl}$) values, which were measured through cyclic voltammetry (CV) method in a non-Faradaic region at different scan rates (Supplementary Figs. 21 and 22). The measured ECSA for Rh/GDY was 295 cm$^2$, larger than that of Rh (167.5 cm$^2$), GDY (34.3 cm$^2$) and CC (26.3 cm$^2$), indicating the largest exposed catalytic sites of Rh/GDY to the reactants (Supplementary Table 8). These results confirm that the Rh/GDY with stepped surfaces has excellent catalytic activity for the production of $H_2$.

Long-term stability is another important criterion for evaluating the use of a catalyst. Experimental results show that there is no significant increase in the applied overpotentials at 100, 500, and 1000 mA cm$^{-2}$ after 8000 continuous cycling voltammetry (CV) cycling cycles (Fig. 4g and Supplementary Fig. 23), and only small decreases in the $j$ of 100 and 500 after 25-h electrolysis (Supplementary Fig. 24). These data demonstrate the excellent stability of Rh/GDY in corrosive saline water conditions. SEM (Supplementary Fig. 25), HAADF-STEM (Supplementary Fig. 26), and STEM-EDS (Supplementary Fig. 27) analysis confirmed that the morphology of the catalyst is well-preserved and all Rh nanocrystals are presented as individual nanoparticle on GDY surface without any aggregation. The C and Rh elements were homogeneous distributed within the whole GDY structure. XPS measurements were performed to determine the chemical valent of Rh/GDY during the cycling tests in alkaline simulated seawater. The C 1s XPS spectra (Supplementary Fig. 28) of Rh/GDY exhibited a positive shift in binding energy, and the Rh 3d XPS spectra (Fig. 4h) exhibited a negative shift in binding energy, suggesting the partial reduction of Rh ions during HER process. XPS survey spectra (Supplementary Fig. 29) of the Rh/GDY measured after the stability tests demonstrated the absence of Na$^+$ and Cl$^-$ on the electrode during the electrocatalysis. As shown in Fig. 4i in the manuscript, the percentages of Rh$^0$ and Rh$^{3+}$ species decreased and the percentage of Rh$^+$ increased, but the catalytic activity of the catalyst remains the same during the whole cycling tests. According to our experimental results, the mixed valence states play a central role in determining the catalytic activity rather than a single valence state. In addition, the Membrane Electrode Assemblies (MEA) experiments were conducted (Supplementary Fig. 30) and confirmed the high stability of our electrocatalysts, which provide a more convincing demonstration of the materials' potential applications.

### Electronic activities and adsorption energetic trends

Density functional theory (DFT) calculations were performed to shed light on the catalytic mechanism developed by Rh/GDY at the atomic level. HAADF-STEM characterizations and experimental results suggest that the formed steps of Rh nanocrystals play the key role in enhancing the catalytic performances. Therefore, a stepped Rh/GDY model (Supplementary Fig. 9) was constructed by loading a three-layer

of 5 × 4 non-periodic Rh atoms with two rows of Rh atoms removed in the outermost layer (50 Rh atoms) on a 2 × 2 periodic supercells of graphdiyne (72 carbon atoms). The difference between the corner and surface sites in stepped Rh/GDY can be clearly identified by the coordination number of Rh atoms (7-coordinated corner atoms and 9-coordinated surface atoms). The HER activities of Rh/GDY and pure Rh were studied by comparing the Gibbs free energies for hydrogen adsorption ($\Delta G_{H^*}$)[71-73]. A highly active HER catalyst must comply with the thermal-neutral requirement ($\Delta G_{H^*} \approx 0$), where the atomic hydrogen binding mode is neither too strong nor too weak to allow concomitant efficient hydrogen adsorption and release. We found that the $\Delta G_{H^*}$ for Rh/GDY (0.02 eV; Supplementary Fig. 31) is much closer to the thermal-neutral state than that of pure Rh (−0.13 eV), which indicate Rh/GDY possesses much better activity than pure Rh. This also highlights the important role of GDY in enhancing the catalytic activity of the catalyst.

In order to deeply understand the reaction mechanism, the whole alkaline HER process of Rh/GDY was then simulated. According to the experimental observations, the HER on stepped Rh/GDY proceeds most likely via a Volmer−Tafel mechanism due to the low Tafel slope of 21 mV dec$^{-1}$. The free energy diagrams for the whole alkaline HER process further confirm that the Volmer-Tafel process is the most preferred among various possible pathways on the stepped Rh/GDY (Fig. 5a). It is found that one $H_2O$ molecule prefers to stably adsorb on the corner site ("*" here refers to the adsorption of intermediates at the corner site) rather than the surface site ("#" here refers to the adsorption of intermediates at the surface site) of the stepped Rh/GDY with a negative adsorption free energy ($\Delta G_{H_2O^*}$) at ambient temperatures. The adsorbed $H_2O^*$ molecule can be then easily dissociated into (H-OH)* that are co-adsorbed on the corner sites. The water dissociation process (i.e., Volmer step) is exothermic with a low energy barrier of 0.76 eV (Fig. 5b), indicating that the stepped Rh/GDY electrocatalyst favors the water dissociation to generate abundant hydroxyl (OH*) and hydrogen (H*) sources. Previous reports show that strongly adsorbed OH* and H* species on the active sites may inhibit the HER process unless they can be immediately released from the catalyst surface. The free energies reflecting the adsorption ability OH* and H* species on the active sites were calculated. It was found that the adsorbed OH*species can be easily removed from the corner site to form an OH$^-$ anion (0.31 eV), which fits well with the widely accepted one-electron reductive desorption mechanism (OH* + e$^-$→ OH$^-$)[74-76]. While the generated hydrogen atoms (H*) are strongly bound to the corner sites with large free-energies of 0.82 and 0.86 eV for Heyrovsky step (H* + $H_2O$ + e$^-$→$H_2^*$ + OH$^-$) and Tafel step (2H* → $H_2^*$), respectively, which inhibit the formation of hydrogen molecule ($H_2$).

Interestingly, we found that the free energies cost for hydrogen formation on the surface site are significantly smaller than those on the corner site. Moreover, the Tafel step on the surface site exhibits much lower free energy barrier (0.19 eV) for hydrogen formation than that of Heyrovsky step (0.39 eV), revealing that the Tafel process is more favorable than the Heyrovsky process on the surface site. These results indicate that the HER via the Volmer−Tafel process is more preferred on the stepped Rh/GDY catalyst, and the corner and surface sites of the stepped Rh/GDY catalyst possess superior thermodynamic activities towards the Volmer and Tafel processes, respectively. If there is a bridge to bring the generated H adsorbed on the corner sites (H*) to the surface site (H#), the stepped Rh/GDY would be an excellent electrocatalyst towards HER. The required energy barrier for the migration of H atom from the * site to the # site is only 0.21 eV on the stepped Rh/GDY (Fig. 5b), indicating that the process is kinetically feasible via the hydrogen spillover process. The Rh/GDY catalyst shows excellent alkaline HER performance via the Volmer−Tafel mechanism (Fig. 5c), which arises from the cooperative effect of corner and surface sites at the highly active stepped Rh crystal face, and the vital role of GDY by inducing-

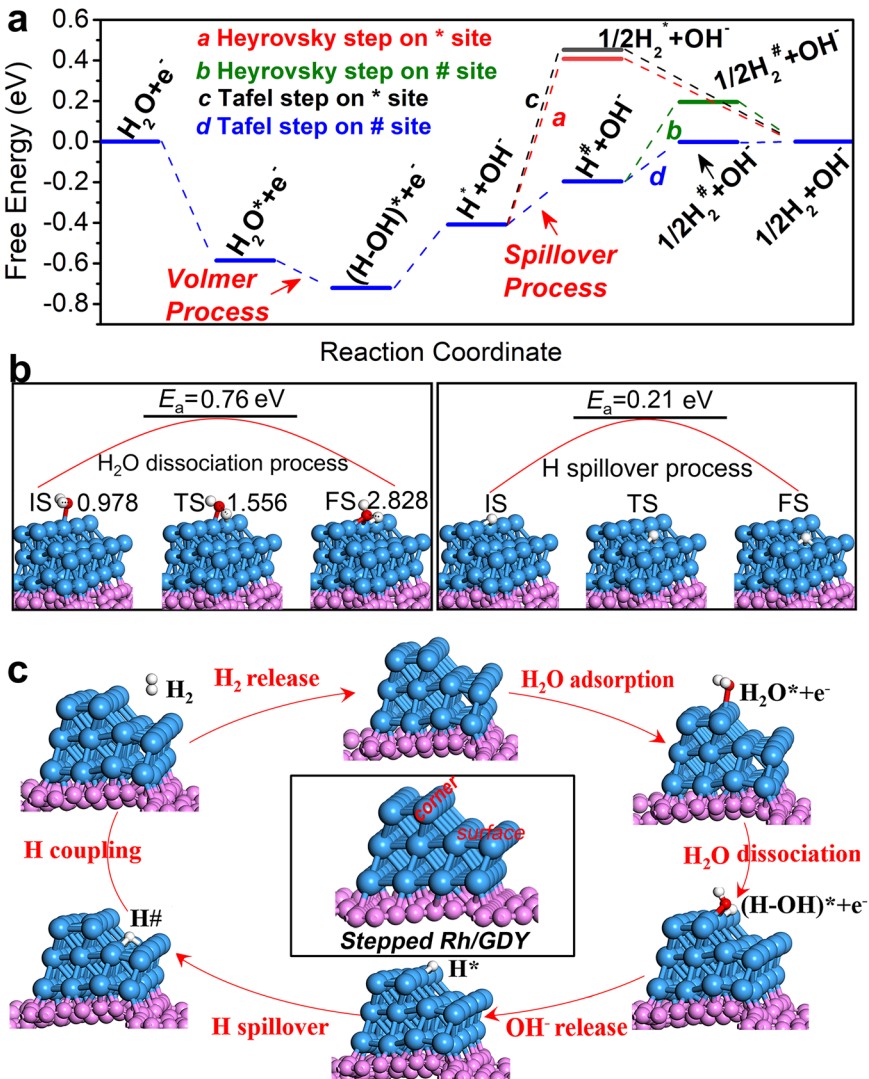

**Fig. 5 | Adsorption energetic trend and pathways on Rh/GDY. a** Energy diagram of Volmer−Tafel pathway for HER on the Rh/GDY. **b** TS transition state (TSs) with the NEB barriers for $H_2O$ dissociation and H migration, respectively. The pink, blue, red, and light gray spheres represent C, Rh, O, and H atoms, respectively. **c** Schematic illustration of the HER reaction mechanisms along the Volmer−Tafel pathway on Rh/GDY.

growth and regulating the electronic structures of stepped Rh metal.

## Discussion

We have demonstrated a facile synthetic method for controllable synthesis of Rh nanocrystals on GDY surface in aqueous solution at low temperatures and ambient pressures. The sample was characterized in detail and a phenomenon of great significance has been found on structure, namely the formation of high-density atomic steps on the surfaces of Rh nanocrystals and $sp$−C-Rh bonds in Rh/GDY. The high-density atomic steps greatly improved the intrinsic properties of the catalysts, leading to the transformative properties of hydrogen production in saline water. The origin of such high catalytic performances of the catalyst was further studied by DFT calculations, which also demonstrated that the exposed high active stepped surfaces and the d-band regulation of the active sites induced by GDY significantly promotes the water dissociation and hydrogen combination processes for HER. This catalyst shows the catalytic performance of HER that we have expected for many years and is currently a promising catalyst for applied on large-scale of hydrogen production with the smallest overpotential value of 65 mV at 1000 mA cm⁻² than all reported catalysts.

## Methods

### Materials
Copper (Cu) foils were cleaned by 1 M HCl, deionized water and acetone, respectively. Carbon fiber cloth (CC) was thoroughly cleaned by nitric acid and deionized water before use.

### Synthesis of porous GDY electrode
Typically, Cu foils (3 cm × 2 cm; 5 pieces) and CC (2 cm × 2 cm; 1 piece) were immersed into 50 mL pyridine at 50 °C for 2 h under Ar. Subsequently, hexaethylbenzene (HEB, 0.6 mg mL⁻¹) pyridine solution was added into the above reactor. The temperature was elevated to 80 °C and kept for 12 h. Then, the CC samples were cleaned by hot DMF, acetone, KOH (4 M), HCl (6 M), KOH (4 M) and deionized water sequentially, and followed by drying in 40 °C vacuum oven for 12 h. The porous GDY samples could then be obtained. In the synthesis process, copper foils were used as the catalyst.

### Synthesis of Rh/GDY
The formic acid (HCOOH, 1 mL) was added to the aqueous solution of $RhCl_3 \cdot xH_2O$ (39%) under stirring at 80 °C. The freshly-prepared GDY electrode was then added. After a 5h-reaction, Rh/GDY was obtained and thoroughly cleaned. The Rh mass loading in Rh/GDY was

determined to be 1.95 $\mu g_{Rh}\ cm^{-2}$ by inductive coupled plasma mass spectrometry (ICP-MS).

## Preparation of Pt/C (20 wt%) electrode

The Pt/C electrode was made by drop-casting 20 $\mu L$ mixed solution (950 $\mu L$ ethanol +50 $\mu L$ 5 wt% Nafion solution) of Pt/C (20 wt%; 1 mg $mL^{-1}$) onto the glass carbon electrode.

## High-density atomic defects characterization

The high-density atomic defects of the samples were determined by the atomically resolved high-resolution high-angle annular dark-field (HAADF) scanning transmission electron microscopy (STEM) and energy dispersive spectroscopy (EDS) imaging on a Talos F200X G2. Individual atoms can be clearly seen.

## Morphological measurements

The morphology of the samples was imaged by using the Hitachi S-4800 field emission scanning electron microscope.

## Composition Characterization

Depth-profiling X-ray photoelectron spectroscopy (XPS) coupled with ion sputtering was conducted (an ion gun is used to etch the catalyst for a constant time to expose a new surface; all binding energies were corrected using carbon element (C 1 s = 284.4 eV). Raman spectra were tested on NT-MDT Spectrum Instruments operating at 473 nm. The power X-ray diffraction (XRD; Cu Kα radiation, $\lambda = 1.54178$ Å) patterns were obtained utilizing the Empyrean X-ray diffractometer. Electron energy-loss spectroscopy (EELS) of the samples were collected using Titan Themis300.

## Electrocatalytic measurements

The electrocatalytic performances of the catalysts were measured by CH Instruments electrochemical workstation with three-electrode system (working electrode: the as-synthesized catalysts; counter electrode: the graphite rod; reference electrode: the saturated calomel electrode (SCE)). The 0.5 M NaCl + 1.0 M KOH alkaline saline aqueous solution was used as a surrogate of seawater, in which the sodium and chlorine ions are dominant compositions. The electrolyte was thoroughly degassed by $H_2$ before recording polarization curves to ensure the accuracy of the measured HER activities. Linear sweep voltammetry (LSV) curves were recorded at 2 mV $s^{-1}$. Cyclic voltammograms (CV) tests were performed in the potential range from −0.9 V to −0.13 V at 100 mV $s^{-1}$. The chronoamperometric tests were conducted at a constant overpotentials to reach the current densities of 100 and 500 mA $cm^{-2}$, respectively. Electrochemical impedance spectroscopy (EIS) was performed in a frequency range from 0.03 to 100,000 Hz at the fixed potential of −0.030 V vs. RHE (IR-non-compensated value) with a signal amplitude perturbation of 5 mV. All potentials were converted to the reversible hydrogen electrode (vs. RHE) according to Eq. 1:

$$E(RHE) = E_{mea} - i \times R_s + pH \times 0.059 + E^0_{SCE} \quad (1)$$

where $E_{mea}$ is the recorded potential, $R$ is the ohmic drop determined by EIS measurements, $i$ is the current, and $E^0_{SCE}$ is 0.242 V.

## Double-layer capacitance ($C_{dl}$) measurement

$C_{dl}$ was measured by using CV method. Briefly, CV curves were recorded in a non-Faradaic range (−0.24 ~ −0.14 V vs. RHE) at 20, 40, 60, 80, 100, 120 and 140 mV $s^{-1}$, respectively. The $C_{dl}$ values for the samples were then calculated by measuring the slope of the fitting line of currents (J) against scan rates. J was calculated according to Eq. 2:

$$J = \frac{J_a - J_c}{2} \quad (2)$$

$J_a$ and $J_c$ represent the anodic and cathodic currents at −0.19 V vs. SCE, respectively.

## Calculation of electrochemically active surface area (ECSA)

The ECSA was calculated according to the Eq. 3:

$$ECSA = C_{dl}/C_s \quad (3)$$

$C_s$ is 40 $\mu F\ cm^{-2}$ based on reported values[77].

## Calculation of the roughness factor ($R_f$)

The $R_f$ was calculated according to the Eq. 4:

$$R_f = C_{dl}/C_s \quad (4)$$

## Calculation of the mass activity

The mass activity ($j_{mass}$) was calculated according to Eq. 5:

$$j_{mass} = \frac{j_{geometrical}}{M_{loading}} \quad (5)$$

where $j_{geometrical}$ is the geometric activity, and $M_{loading}$ is the catalyst loading per geometric surface area. For Rh/GDY, the $M_{loading}$ is 1.95 $\mu g\ cm^{-2}$.

## Calculation of turn over frequency (TOF) and number of active sites

The TOF values can be obtained according to Eq. 6:

$$TOF = \frac{j}{2 * F * n_s} = \frac{j * N_A}{2 * F * N_s} = \frac{j * N_A}{2 * F * N_{s,flat} * R_f} \quad (6)$$

in which j, F, $N_A$, and $R_f$ are the current density, Faraday constant (96485.3 C $mol^{-1}$), Avogadro's number (6.022 × $10^{23}\ mol^{-1}$), and roughness factor, respectively. ns, $N_s$, and $N_{s,flat}$ stand for the number of moles of active sites per geometric surface area, the number of active sites per geometric surface area, and the number of surface sites per 1 $cm^2$ of the flat standard electrode (2 × $10^{15}\ cm^{-2}$ based on previous results[77]), respectively.

## Calculation Details

DFT calculations in the study were performed using Vienna ab-initio simulation package (VASP). To describe the exchange-correlation functional, the Perdew-Burke-Ernzerbof (PBE) functional within the generalized-gradient approximation (GGA) was chosen[78]. To describe the core-valence electron interaction, the Blöchl's all-electron-like projector augmented wave (PAW) pseudo-potential was used[79]. For the structural optimizations, the plane wave basis sets with a kinetic cutoff energy of 450 eV, and the Monkhost-Pack grid with a k-point meshes of 3 × 3 × 1 were applied[80]. For the density of states (DOS) calculations, the much closer k-point meshes of 9 × 9 × 1 were employed. The $d$-band centers of metal atoms were obtained by evaluating the centroid of the projected DOS relative to Fermi level. During the structural optimization, all atoms were allowed to be relaxed until the energy is less than 1.0 × $10^{-6}$ eV/atom, and the ionic force is less than 0.01 eV/Å. The van der Waals (vdW) interaction was considered by using the Grimmer's DFT-D3 method[81]. A continuum solvent model[82] is employed to consider the solvation effect, which has been applied in previous studies for hydrogen evolution[71]. To avoid the periodic interactions between two slabs, a vacuum layer as large as 20 Å was used along the c direction. The climbing image nudged elastic band (CI-NEB) method was used to search the transition states structures[83]. There is only one virtual frequency for all the calculated transition states.

## Structural models

HAADF-STEM characterizations and experimental results suggest that the formed steps of Rh nanocrystals play the key role in enhancing the catalytic performances. Therefore, a stepped Rh/GDY model (Supplementary Fig. 9) was constructed by loading a three-layer of 5 × 4 non-periodic Rh atoms with two rows of Rh atoms removed in the outermost layer (50 Rh atoms) on a 2 × 2 periodic supercells of graphdiyne (72 carbon atoms). The difference between the corner and surface sites in stepped Rh/GDY can be clearly identified by the coordination number of Rh atoms (7-coordinated corner atoms and 9-coordinated surface atoms). The distance between the two nearest Rh cluster is 10.753 Å (Supplementary Fig. 9), which is large enough to avoid the interaction. To avoid the lattice mismatch between Rh nanocrystal and GDY, the periodicity of Rh was destroyed, while the periodicity of GDY was remained in the study.

## Calculation method for Gibbs free energy

The computational hydrogen electrode model[84] was utilized to study the Gibbs free energies of adsorbed intermediates on catalysts surface involved in the alkaline hydrogen evolution process ($\Delta G_{ads}$ determined by $G_{adsorbate/surface} - G_{surface} - G_{adsorbate}$; the $G_{adsorbate/surface}$, $G_{surface}$ and $G_{adsorbate}$ are the Gibbs free energies of catalyst surface with adsorbate, bare catalyst surface and bare adsorbate, respectively. The terms of $G_{adsorbate/surface}$, $G_{surface}$ and $G_{adsorbate}$ were calculated by adding the zero-point energy correction ($\Delta$ZPE) and the entropic correction (T$\Delta$S, 298 K adopted in this work). The OH* desorption step in alkaline electrolytes can be expressed by the widely reported one-electron reductive desorption mechanism: OH* + e$^-$ → OH$^-$(aq) + *. The Gibbs free energy of OH$^-$ is equivalent to the energy difference of $G_{H2O}-1/2G_{H2}$ in the study, where $G_{H2O}$ and $G_{H2}$ are the chemical potentials of liquid H$_2$O and gas H$_2$, respectively (taken from the NIST WebBook).

## Data availability

All relevant data that support the findings of this study are available from the corresponding author upon reasonable request. The Source data generated in this study are provided in the Source Data file. Source data are provided with this paper.

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

## Acknowledgements

This work was supported by the National Nature Science Foundation of China (21790050, 21790051, 22021002, 22005310), the National Key Research and Development Project of China (2018YFA0703501), the Key Program of the Chinese Academy of Sciences (XDPB13), and the Post-doctoral Science Foundation of China (2019M660806).

## Author contributions

Y.L. conceived and designed the research, and critically revised the manuscript. Y.X. and Y.G. designed the experiments. Y.G. synthesized the catalysts, carried out the experiments, analyzed the data, and wrote the draft. Y.X. helped the data analysis, and organized and revised the draft. L.Q., C.X. and X.Z. gave useful help during the experiments. F.H. performed the theoretical calculations.

## Competing interests

The authors declare no competing interests.
