## [Peer Review File · Nature Communications]

Rhodium nanocrystals on porous graphdiyne for electrocatalytic hydrogen evolution from saline waterREVIEWER COMMENTS

Reviewer #1 (Remarks to the Author):

The authors reported the Rh/GDY for high-performance conversion of seawater to hydrogen. However, the HER of single metal or metal clusters on GDY based catalysts had been widely investigated. I didn't see any novelty of this work. I'm unable to recommend it for publication on Nat. Comm. The following are some comments for the calculation parts that hope to be helpful for the authors:

- 1) For the computational methods, the authors used very simple parameters and the details of the model were not provided which makes the results not convinced. Such as they didn't take into account of the long-range dispersion interactions between the adsorbates and catalysts, the convergence criteria were rough, the K-points was too small for optimization let alone the electronic calculations.
- 2) The important solvent effect hasn't been taken into consideration. I highly recommend the explicit model when consider the solvent effect. In this case, the Tafel process and Heyrovsky process can be well compared.
- 3) The size of the model should be carefully checked, such as the two nearest Rh cluster should be far enough to avoid the interaction.
- 4) As the authors mentioned in the introduction parts, "A variety of porous GDY based catalysts have been developed, e.g., GDY-based zero-valent atomic catalysts³⁰⁻³², heterostructured catalysts³³⁻³⁵, quantum dots catalysts³⁶⁻³⁹, and metal-free catalysts", the metal doped GDY-based HER catalysts had been extensively reported. The C atoms of GDY could also serve as the HER active sites when loading metal atoms. How to verify the real active sites?
- 5) The authors pointed that the stepped surface and special interactions form between GDY and Rh endowed the 71 catalyst with excellent electrocatalytic activity and stability. I didn't see any evidence of the role of GDY. Could the high activity origin from the stepped Rh? The authors at least should compare the results with the bare stepped Rh.
- 6) For the comparison of Tafel process and Heyrovsky process, the NEB barriers should be calculated.

Reviewer #2 (Remarks to the Author):

High current density HER and OER is key for industrial water splitting applications. In a more general sense, high current density reactions are prerequisites to make any electrochemical conversion practically useful. In this work, Gao et al. have reported a Rh nanocluster/graphdiyne (GDY) catalyst which can electrochemically split sea water. This catalyst only needs a very small overpotential of 65 mV to reach a high current density of 1000 mA/cm². This performance is impressive in terms of sea water splitting, and is among the very top performance. The work thus contains novel elements and important merits. Comprehensive experimental and theoretical studies were performed carefully which make the most conclusions solid and valid. The work was also well organized and written. I am happy to recommend its publication after properly addressing the following questions.

1. What are the key and general requirements to catalysts to achieve high current density HER? Please discuss this point as it may inspire future research. In addition, recently, there are quite a lot of progress in sea water splitting which may be included in the revised version, e.g., PNAS, 2019; 116 (14): 6624. Advanced Functional Materials 2021, 31, 2010367.
2. Line 125, the authors wrote that "HAADF-STEM images reveal that the edges of (111) planes are not straight but contain numerous of mono-layered (Fig. 2f) and multi-layered steps (Fig. 2g-2j)." This is an interesting observation and feature of catalysts. Could you please add more discussions on the reason to form such steps on Rh clusters? I think this is an interesting point.
3. How did the authors determine the area to calculate the current density? Is it the electrode area of

the ECSA-normalized area? Please make this point clear. In Figure 4c, as the current density is pretty high, the term "Tafel slope" is not suitable here as it only refers to small currents regime (<1-100 mA/cm²). The authors may refer to the ratio defined in this work to discuss this point more precisely (Nat Commun 2019, 10, 269.).

4. In Figure 4g, please convert 8000 cycles into time/hours. In addition, actually Figure S22 is the more common way to show the durability of catalyst. It is also suggested to include LSV curves without iR-compensation maybe in SI, and specify how the compensation was done. These may make the work more clear in terms of presentation.

5. Some other minor issues. a). The video is impressive but is too long, and is suggested to speed up by ~10 times. b). reference numbering seems to be not correct in sequence. Please double check. c). The quality of several figures can be further improved to make them clear, e.g., Figure 4.

Reviewer #3 (Remarks to the Author):

The paper by Gao et al. shows a very interesting material for electrocatalytic hydrogen production in alkaline conditions. This work shows that the combination of rhodium nanoparticles and graphdiyne provides large surface and active area for HER due to the synergistic effect of the two components. The authors have done a very nice job combining state of the art techniques for the visualization of the Rh nanoparticles, x-ray spectroscopy and computational work, all to unravel the reaction mechanism. This shows that the stepped surfaces produced as a result of the truncated octahedrons play a very significant role in the reaction, in particular on the first step of water adsorption and dissociation.

There are a number of aspects that the authors should consider before publication:

1) The use of the word seawater should be removed. I understand it is becoming standard practice in the community, but experiments are done most of the times using a NaCl solution in alkaline conditions. The whole manuscript should be amended using the term saline electrolysis. This has a significant effect, since in real seawater HER would be mostly affected by the precipitation of species that are not present when using 1M KOH + 0.5M NaCl. Likewise, as it is shown by the authors, the addition of NaCl (half concentration from KOH) does not have an impact on the overall performance of the cell.

2) References in the introduction should be checked. Ref 1-7 have little to do with the sentence, similarly for 8-12. Authors should check recent literature for saline water electrolysis. Same for ref 14-19, the sentence related to Rh catalysts, whereas most of the referenced articles are based on Pd or Pt.

3) An important aspect in the paper is to specify that HER is the focus. The literature has publications on OER, HER or complete saline water electrolysis, and this paper only focuses on HER. This should be more clear as there are sentences that cannot be applied to OER (line 51, carbon materials are considered good catalysts... this cannot be applied to OER due to corrosion of the material).

4) Line 60 states "graphdiyne as a catalyst with excellent..", authors should specify the reactions where the material is a good catalyst.

5) Figure 1 shows very nicely the electrodes and electron microscopy images of the materials. The histogram should have the number of particles counted.

6) The comparison of performance with Rh NPs and different carbon substrates is nice, but authors should check that the amount of Rh is the same with the different substrates.

7) HAADF-STEM in Fig 2 is impressive, you can clearly see the stepped surface which is crucial for the high activity. Has this analysis been conducted for different particles? Considering the different size (as seen in histogram), does this have an effect on the "relative density" of steps among different particles?

8) The methodology for electrochemical measurements should be checked. There is missing information on some measurements, and it says to use Pt/C as a WE instead of the prepared materials. Authors should justify the use of H₂-saturated electrolyte too. Is the counter electrode a graphite rod?

9) The comparison with other catalysts in Fig 4 and SI is very useful, has this been done using the same experimental conditions? I note that in SI there is the table separating purely 1M KOH from 1M KOH + saline electrolyte (or real seawater).

10) The discussion of TOF (line 210) should be amended. The comparison between exchange current density 1.3 vs 0.65 is not two orders of magnitude higher, please rephrase accordingly.

11) The analysis of the electrodes post-electrocatalysis indeed shows good stability, however EM images do show a degree of aggregation and the resolution of the images is much worse compared to the one before catalysis. Some effort should be placed in improving the images and discuss the results.

12) Also in analysis, have XPS measurements shown the presence of Na⁺ or Cl⁻ on the electrode?

13) In DFT section, it is very interesting the different adsorption energy between corner and surface, even for water molecule. Is there any change on partial charge density that could give a justification for such an observation?

14) Note that for the experimental procedure, copper foils are added with a piece of CC. This is not mention in the manuscript text and I wonder if there is some Cu on the surface of the electrode as well? Which is the role of copper foils in the synthesis process?

15) The text should be checked, there are typos and errors to be amended, including repetition of a sentence in lines 159-161.

Reviewer #4 (Remarks to the Author):

Gao et al. reported Rhodium (Rh) nanocrystals on porous graphdiyne (GDY) as hydrogen evolution electrocatalysts in electrochemical seawater splitting. The electrocatalyst exhibits a record low overpotential of 65 mV to achieve a current density of 1 A/cm². Although the results are very interesting, the discussions are not convincing. Please see the detailed comments below.

1. There lack details of GDY synthesis: It looks like carbon cloth is the substrate of GDY. What is the role of copper foils? Have the authors tried different experimental parameters (other than the reported procedure) to optimize GDY?

2. Line 95, the authors report a very narrow size distribution (with a distribution of 0.03 nm). This does not agree with the histogram in Fig. 1a. Please clarify.

3. Lin 140, the authors claim that "there are no changes in the compositions and chemical states". However, the surface peaks of sample 1 are wider and show higher binding energy. Please discuss the

reason.

4. Line 153, "due to the reducibility of GDY...", "the Rh ions can be easily reduced to metallic species". But then why GDY can oxidize Rh up to Rh³⁺?

5. Line 157, what are the "typical semiconductor properties"? Does it indicate a bandgap and low electrical conductivity of the mixed-valence states?

6. The calculation shows that the Volmer–Tafel process is likely the reaction path. However, a ~0.20 eV energy barrier exhibits in the Tafel process. With such a barrier, please discuss how can the Rh/GDY drive the 1 A/cm² current density at 65 mV?

REVIEWER COMMENTS

Reviewer #1 (Remarks to the Author):

The authors reported the Rh/GDY for high-performance conversion of seawater to hydrogen. However, the HER of single metal or metal clusters on GDY based catalysts had been widely investigated. I didn't see any novelty of this work. I'm unable to recommend it for publication on Nat. Comm. The following are some comments for the calculation parts that hope to be helpful for the authors:

Response: Directly producing green hydrogen (H₂) on large-scale from seawater electrolysis is a long-sought dream of mankind. The challenge is the lack of a catalyst to realize efficient seawater splitting for large-scale H₂ production with large current densities at low overpotentials that can meet the industrial requirements. This is also a key step to achieve zero pollution and hydrogen fuel highly economy. However, the traditional catalysts pose significant trouble for the efficient large-scale H₂ production, such as large overpotential and low long-term stability at high current densities.

What we want to show is that we cannot ignore innovative research and explore new concepts of catalysts in the field of catalysis. We can confidently answer the reviewer that our research is very original in both methodology and concept, such as (i) we propose the new concept of high-density atomic defect, and the high-density atomic defects was observed on the catalyst; (ii) we show a new phenomenon of the incomplete charge transfer between metal quantum dots and sp-/sp²-hybridized carbon, and greatly enhance the intrinsic activity of the catalyst; and (iii) An advanced method was designed to realize the controlled synthesis of metal Rh quantum dots with high-density step distribution on GDY. The advanced concept, method and phenomenon have been first reported in the manuscript. Our results also demonstrate that the most efficient conversion of seawater to hydrogen fuel and sets a new record with 1000 mA cm⁻² at very small overpotential of 65 mV at room temperatures and ambient pressures.

1) For the computational methods, the authors used very simple parameters and the details of the model were not provided which makes the results not convinced. Such as they didn't take into account of the long-range dispersion interactions between the adsorbates and catalysts, the convergence criteria were rough, the K-points was too small for optimization let alone the electronic calculations.

Response: Actually, we have taken into account of these parameters and tested them in the system.

Take the vital step of H adsorption during HER as an example, the results of the calculated H adsorption energies using different criteria are shown in Fig. R1. It was observed that the higher convergence accuracy (i.e., a cutoff energy of 450 eV, an energy tolerance of 1.0×10^{-6} eV/atom and a force tolerance of 0.01 eV/Å) and the more mesh K-points (i.e., $3 \times 3 \times 1$) have no influences on the adsorption energies. The van der Waals long-range dispersion interaction is further considered using the DFT-D3 method for comparison, which also almost have no influence on the adsorption energies. According to the reviewer's suggestion, we perform all the DFT calculations again using the above-mentioned new parameters with higher criteria in the revised manuscript. Specially, the much mesh $9 \times 9 \times 1$ K-point grid is employed for the electronic structure calculations.

Fig. R1. Comparison of adsorption energies for H adsorbate on stepped Rh/GDY surface with different criteria.

2) The important solvent effect hasn't been taken into consideration. I highly recommend the explicit model when consider the solvent effect. In this case, the Tafel process and Heyrovsky process can be well compared.

Response: A continuum solvent model (*Chem. Rev.* 2005, 105, 2999–3094) is employed to consider the solvation effect, which has been applied in previous studies for hydrogen evolution (*Nat. Commun.* 2019, 10, 149). As shown in Fig. R2, the solvent effect does not affect the HER reaction mechanisms and the conclusions in this study.

Fig. R2. Free energy diagrams for various possible pathways towards HER on the stepped Rh/GDY in (a) previous version and (b) revised version.

3) The size of the model should be carefully checked, such as the two nearest Rh cluster should be far enough to avoid the interaction.

Response: The distance between the two nearest Rh cluster is 10.753 Å (Fig. R3), which is large enough to avoid the interaction.

Fig. R3. The periodic supercell model of stepped Rh/GDY.

4) As the authors mentioned in the introduction parts, “A variety of porous GDY based catalysts have been developed, e.g., GDY-based zero-valent atomic catalysts³⁰⁻³², heterostructured catalysts³³⁻³⁵, quantum dots catalysts³⁶⁻³⁹, and metal-free catalysts”, the metal doped GDY-based HER catalysts had been extensively reported. The C atoms of GDY could also serve as the HER active

sites when loading metal atoms. How to verify the real active sites?

Response: It has been proved theoretically and experimentally that the surface structure of GDY is composed of highly distributed alkyne bonds, in which the π bond is very active and has strong interactions with metals and heteroatoms, as well as incomplete charge transfer effect, and the uneven surface charge distribution on the surface of GDY caused by the co-hybridization of sp and sp^2 carbon are the natural active sites (e.g., *Chem. Rev.* 2018, 118, 7744-7803; *Chem. Soc. Rev.* 2022, 51, 2681-2709; *Nat. Commun.* 2018, 9, 1460; *J. Am. Chem. Soc.* 2022, 144, 1921-1928; etc.).

In addition, our calculation results also verify the real active sites of our electrocatalysts in this work. Gibbs free energy for hydrogen adsorption (ΔG_{H^*}) is a generally acknowledged descriptor to determine the HER activity and verify the real active sites of a catalyst. A highly active HER sites of catalyst must comply with the thermal-neutral requirement ($\Delta G_{H^*} \approx 0$), where the atomic hydrogen binding mode is neither too strong nor too weak to allow concomitant efficient hydrogen adsorption and release. As shown in Fig. R4, the calculated ΔG_{H^*} value for stepped Rh/GDY (0.02 eV) is much closer to the thermal-neutral state compared to that for pure Rh ($\Delta G_{H^*} = -0.13$ eV), highlighting the important role of GDY in enhancing the catalytic activity of the electrocatalyst. Besides, the ΔG_{H^*} value for different C sites in stepped Rh/GDY are much far away from the thermal-neutral state, thereby are not the HER active sites in this system.

Fig. R4. Calculated ΔG_{H^*} values for Rh and different sites in stepped Rh/GDY.

5) The authors pointed that the stepped surface and special interactions form between GDY and Rh endowed the catalyst with excellent electrocatalytic activity and stability. I didn't see any evidence of the role of GDY. Could the high activity origin from the stepped Rh? The authors at least should compare the results with the bare stepped Rh.

Response: As we have discussed in the manuscript, the excellent activity of the catalyst is indeed

mainly originated from the stepped Rh. However, the stepped Rh can only be obtained by using GDY as the growing substrate, strongly evidencing the decisive role of GDY in inducing the controlled growth of stepped Rh. Accordingly, it is meaningless to compare the results with the bare stepped Rh without GDY. Besides, GDY can efficiently regulate the electronic structures of stepped Rh metal (Fig. R5 and Fig. R6) that endow it excellent electrocatalytic activity and stability, also verifying the vital role of GDY.

Fig. R5. The charge density difference of Rh/GDY.

Fig. R6. Comparison of projected density of state (PDOS) with *d*-center of different sites in Rh/GDY and Rh.

6) For the comparison of Tafel process and Heyrovsky process, the NEB barriers should be calculated.

Response: It is hard to accurately calculate the barrier for the combination of ($H^+ + e^-$) pair with adsorbed H^* in the Heyrovsky step using the NEB method. Instead, we have compared the free energy barriers of Tafel process and Heyrovsky process (Fig. R2), where the free energy of ($H^+ + e^-$) at standard conditions was assumed as the energy of $1/2H_2$. The calculated barriers for the Tafel process

and Heyrovsky process are 0.19 eV and 0.39 eV, respectively. Meanwhile, the transition states barriers for H₂O dissociation and H migration have been calculated using the NEB method (Fig. R7).

Fig. R7. Transition states (TSs) with the NEB barriers for H₂O dissociation and H migration, respectively. The pink, blue, red, and white spheres represent C, Rh, O, and H atoms, respectively.

Reviewer #2 (Remarks to the Author):

High current density HER and OER is key for industrial water splitting applications. In a more general sense, high current density reactions are prerequisites to make any electrochemical conversion practically useful. In this work, Gao et al. have reported a Rh nanocluster/graphdiyne (GDY) catalyst which can electrochemically split seawater. This catalyst only needs a very small overpotential of 65 mV to reach a high current density of 1000 mA/cm². This performance is impressive in terms of seawater splitting, and is among the very top performance. The work thus contains novel elements and important merits. Comprehensive experimental and theoretical studies were performed carefully which make the most conclusions solid and valid. The work was also well organized and written. I am happy to recommend its publication after properly addressing the following questions.

Response: Thanks for the positive comments and helpful suggestions.

1. What are the key and general requirements to catalysts to achieve high current density HER? Please discuss this point as it may inspire future research. In addition, recently, there are quite a lot of progress in sea water splitting which may be included in the revised version, e.g., PNAS, 2019; 116 (14): 6624. Advanced Functional Materials 2021, 31, 2010367.

Response: Thanks for your helpful suggestion.

Discussion on the key and general requirements to achieve high current density HER have been added in the revised manuscript as follow:

From the further perspective of mass hydrogen production through water splitting, the electrocatalysts should have the following necessary properties: (i) high intrinsic catalytic activity that can achieve high current density HER at low overpotentials; (ii) strong corrosion resistance and

electrochemical/mechanical stability at large current densities; (iii) efficient gas/mass diffusion ability capable of separating the formed H₂ bubbles to maintain the catalytic activity for high current density HER; and (iv) low cost.

These important references have also been cited in the revised manuscript.

2. Line 125, the authors wrote that “HAADF-STEM images reveal that the edges of (111) planes are not straight but contain numerous of mono-layered (Fig. 2f) and multi-layered steps (Fig. 2g-2j).” This is an interesting observation and feature of catalysts. Could you please add more discussions on the reason to form such steps on Rh clusters? I think this is an interesting point.

Response: Thanks for your valuable suggestion.

Experimental results indicate that the GDY has high affinity with Rh atoms which facilitates the selective and stable anchoring of Rh atoms. The high intrinsic reductivity of GDY controllably regulates the interface of Rh growth and guides the formation of octahedron. The stepped morphologies might then be produced as a result of the truncated octahedrons.

Corresponding discussion have been updated in the revised manuscript.

3. How did the authors determine the area to calculate the current density? Is it the electrode area of the ECSA-normalized area? Please make this point clear. In Figure 4c, as the current density is pretty high, the term “Tafel slope” is not suitable here as it only refers to small currents regime (<1-100 mA/cm²). The authors may refer to the ratio defined in this work to discuss this point more precisely (Nat Commun 2019, 10, 269.).

Response: Thanks for your suggestion.

The geometric surface area was used to calculate the current density. The current density (j ; mA cm⁻²) is the current (mA) at a given overpotential η (mV) per geometric surface area (cm²).

According to previous work (Nat. Commun. 2019, 10, 269), we calculated the ratios of overpotential to current density ($\Delta\eta/\Delta\log|j|$, $R_{\eta/j}$) for Rh/GDY in different current density ranges to evaluate the performance of a catalyst at high current densities more precisely. As shown in Fig. R8, the ratio for Rh/GDY remains smaller (< 80 mV dec⁻¹) with the increase of the current density.

This valuable reference has been cited in the revised manuscript.

Corresponding description have been provided in the revised manuscript and Supporting Information.

Fig. R8. Ratios of $\Delta\eta/\Delta\log|j|$ ($R_{\eta/j}$) for Rh/GDY in different current density ranges, which can be used as an indicator to evaluate the performance of a catalyst at high current densities.

4. In Figure 4g, please convert 8000 cycles into time/hours. In addition, actually Figure S22 is the more common way to show the durability of catalyst. It is also suggested to included LSV curves without iR -compensation maybe in SI, and specify how the compensation was done. These may make the work more clear in terms of presentation.

Response: Thanks for your suggestion.

The x-coordinate for the Inset of Figure 4g was converted into hours and shown in Fig. R9.

The LSV curves without iR -compensation have also been provided in the revised Supplementary Materials, as shown in Fig. R10 here.

The final potentials have been converted to the reversible hydrogen electrode (vs. RHE) by iR compensation according to the following formulas:

$$E(\text{RHE}) = E_{\text{measure}} - iR_s + 0.059 * \text{pH} + E^0(\text{SCE})$$

where E_{measure} is the measured potential, i is the current, $E^0(\text{SCE})$ is 0.242 V, and R is the uncompensated resistance as determined by electrochemical impedance spectroscopy (EIS).

These results have also been provided in the revised manuscript and Supplementary Materials, respectively.

Fig. R9. Durability tests of Rh/GDY over 31 hours at high current densities of 100, 500 and 1000 mA cm⁻², respectively.

Fig. R10. The LSV polarization curves without *iR*-compensation of the Rh/GDY.

5. Some other minor issues. a). The video is impressive but is too long, and is suggested to speed up by ~10 times. b). reference numbering seems to be not correct in sequence. Please double check. c). The quality of several figures can be further improved to make them clear, e.g., Figure 4.

Response: Thanks for your valuable suggestion.

a) The video has been sped up by ~10 times.

b) We have checked this carefully and numbered the references in sequence.

c) The quality of the figures has been improved.

All these issues have been updated in the revised manuscript.

Reviewer #3 (Remarks to the Author):

The paper by Gao et al. shows a very interesting material for electrocatalytic hydrogen production in alkaline conditions. This work shows that the combination of rhodium nanoparticles and graphdiyne provides large surface and active area for HER due to the synergistic effect of the two components. The authors have done a very nice job combining state of the art techniques for the visualization of the Rh nanoparticles, x-ray spectroscopy and computational work, all to unravel the reaction mechanism. This shows that the stepped surfaces produced as a result of the truncated octahedrons play a very significant role in the reaction, in particular on the first step of water adsorption and dissociation.

There are a number of aspects that the authors should consider before publication:

Response: We thank the reviewer for the positive comments and helpful suggestions.

1) The use of the word seawater should be removed. I understand it is becoming standard practice in the community, but experiments are done most of the times using a NaCl solution in alkaline conditions. The whole manuscript should be amended using the term saline electrolysis. This has a significant effect, since in real seawater HER would be mostly affected by the precipitation of species that are not present when using 1M KOH + 0.5M NaCl. Likewise, as it is shown by the authors, the addition of NaCl (half concentration from KOH) does not have an impact on the overall performance of the cell.

Response: Thanks for your suggestion. We have changed “seawater” to “saline water” throughout the manuscript where applicable.

2) References in the introduction should be checked. Ref 1-7 have little to do with the sentence, similarly for 8-12. Authors should check recent literature for saline water electrolysis. Same for ref 14-19, the sentence related to Rh catalysts, whereas most of the referenced articles are based on Pd or Pt.

Response: Thanks for your kind reminding. We have carefully checked and revised these issues in the revised manuscript.

3) An important aspect in the paper is to specify that HER is the focus. The literature has publications on OER, HER or complete saline water electrolysis, and this paper only focuses on HER. This should

be more clear as there are sentences that cannot be applied to OER (line 51, carbon materials are considered good catalysts... this cannot be applied to OER due to corrosion of the material).

Response: We have carefully checked and updated the cited references mainly focused on HER. The cited references for the sentences in line 51 (As an important traditional metal support, carbon materials show great application value in catalysis field, therefore, carbon materials are considered as the key component of high-performance catalysts) have also updated in the revised manuscript.

4) Line 60 states "graphdiyne as a catalyst with excellent..", authors should specify the reactions where the material is a good catalyst.

Response: This has been updated in the revised manuscript as follow:

These natural advantages make GDY as a catalyst with excellent comprehensive performance for hydrogen evolution reaction (HER)³⁰⁻³⁵, oxygen reduction reaction (ORR)^{36,42}, oxygen evolution reaction (OER)^{34,35}, overall water splitting (OWS)^{34,35}, nitrogen reduction reaction (NRR)^{37,38}, CO₂ reduction reaction (CO₂RR)³⁹, methanol oxidation reaction (MOR)⁴⁰ and CO catalytic oxidation⁴¹.

5) Figure 1 shows very nicely the electrodes and electron microscopy images of the materials. The histogram should have the number of particles counted.

Response: The number of particles counted have been provided in Figure 1 in the revised manuscript.

6) The comparison of performance with Rh NPs and different carbon substrates is nice, but authors should check that the amount of Rh is the same with the different substrates.

Response: The amounts of Rh for different samples were determined by inductive coupled plasma mass spectrometry (ICP-MS) method. The results show that the Rh mass loading for Rh NPs and those loaded on different carbon substrates are almost identical.

7) HAADF-STEM in Fig. 2 is impressive, you can clearly see the stepped surface which is crucial for the high activity. Has this analysis been conducted for different particles? Considering the different size (as seen in histogram), does this have an effect on the "relative density" of steps among different particles?

Response: Thank you for your concern. Rh nanocrystals with different sizes have also been analyzed (Fig. R11). The stepped surface could be observed from all these Rh nanocrystals.

From current experimental results, it is hard to determine whether the particle size has an effect on the "relative density" of steps.

Fig. R11. Rh nanocrystals with different sizes.

8) The methodology for electrochemical measurements should be checked. There is missing information on some measurements, and it says to use Pt/C as a WE instead of the prepared materials. Authors should justify the use of H₂-saturated electrolyte too. Is the counter electrode a graphite rod?

Response: We have checked the methodology for electrochemical measurements and added the missing information in the revised manuscript.

In the electrochemical measurements, the as-prepared materials were used as the working electrode (WE). For comparison, the Pt/C was used as a WE instead of the prepared materials. The graphite rod was used as the counter electrode.

The H₂-saturated electrolyte were used for each electrochemical measurement to ensure the accuracy of the measured HER activities. The reason is the following: for the measurement of HER kinetics, the potential of the working electrode undergoes a Nernst-shift depending on the H₂-partial pressure ($p(H_2)$) in its vicinity; and the electrolyte should be saturated with H₂ to have a constant backpressure. Therefore, we measured HER under H₂-saturated condition.

9) The comparison with other catalysts in Fig 4 and SI is very useful, has this been done using the same experimental conditions? I note that in SI there is the table separating purely 1 M KOH from 1 M KOH + saline electrolyte (or real seawater).

Response: The performances of the as-synthesized catalysts in this work were measured in 1.0 M KOH + 0.5 M NaCl aqueous solutions. To provide more information to readers, we compared the performances of our catalysts with those tested in strong alkaline conditions including the purely 1 M KOH, 1 M KOH + 0.5 M NaCl electrolyte, and the real seawater.

10) The discussion of TOF (line 210) should be amended. The comparison between exchange current density 1.3 vs 0.65 is not two orders of magnitude higher, please rephrase accordingly.

Response: Thanks for your kind reminding. This has been revised in the revised manuscript as follow. *Rh/GDY exhibits a j_0 of 1.3 mA cm^{-2} , which is higher than that of the reported HER electrocatalysts in alkaline conditions such as RhPd-H/C (0.65 mA cm^{-2})⁵⁰, $\text{W}_1\text{Mo}_1\text{-NG}$ (0.26 mA cm^{-2})⁵¹, and $\text{ES-WC/W}_2\text{C}$ (0.58 mA cm^{-2})⁵² (Supplementary Fig. 17 and table 4).*

11) The analysis of the electrodes post-electrocatalysis indeed shows good stability, however TEM images do show a degree of aggregation and the resolution of the images is much worse compared to the one before catalysis. Some effort should be placed in improving the images and discuss the results.

Response: Thanks for your suggestion. The TEM images with higher resolution and discussion have been provided in the revised manuscript.

12) Also in analysis, have XPS measurements shown the presence of Na⁺ or Cl⁻ on the electrode?

Response: XPS survey spectra of the Rh/GDY measured after the stability tests were shown in Fig. R12. There no signals corresponding to the Na⁺ and Cl⁻ species could be observed. These results demonstrated the absence of Na⁺ and Cl⁻ on the electrode.

Fig. R12. The XPS survey spectra of Rh/GDY obtained after stability tests.

13) In DFT section, it is very interesting the different adsorption energy between corner and surface, even for water molecule. Is there any change on partial charge density that could give a justification for such an observation?

Response: Thanks for your valuable suggestion. We have compared the charge density difference of H₂O adsorption on the corner site and surface site in Rh/GDY, respectively. As shown in Fig. R13, the charge transfer between H₂O and the corner site is more obvious than the surface site, which indicates that the water molecule has much stronger interaction with the corner site than the surface site.

Fig. R13 Comparison of charge density difference of H₂O adsorption on the corner site (left) and surface site (right), respectively. Isosurface=0.001 e/Bohr³.

14) Note that for the experimental procedure, copper foils are added with a piece of CC. This is not mention in the manuscript text and I wonder if there is some Cu on the surface of the electrode as well? Which is the role of copper foils in the synthesis process?

Response: Thanks for your concern.

There is no any Cu species on the surface of the electrode, as evidenced by the high-resolution XPS analysis (Fig. R14) and ICP-MS results.

In the synthesis process, coppers foils were used as the catalyst.

Corresponding discussion have also been added in the revised manuscript.

Fig. R14. The XPS survey spectra of (a) GDY and (b) Rh/GDY and Rh.

15) The text should be checked, there are typos and errors to be amended, including repetition of a sentence in lines 159-161.

Response: Thanks. We have carefully checked the text. The typos and errors have been corrected in the revised manuscript.

Reviewer #4 (Remarks to the Author):

Gao et al. reported Rhodium (Rh) nanocrystals on porous graphdiyne (GDY) as hydrogen evolution electrocatalysts in electrochemical seawater splitting. The electrocatalyst exhibits a record low overpotential of 65 mV to achieve a current density of 1 A/cm². Although the results are very interesting, the discussions are not convincing. Please see the detailed comments below.

Response: Thanks for your comments and valuable suggestions.

1. There lack details of GDY synthesis: It looks like carbon cloth is the substrate of GDY. What is the role of copper foils? Have the authors tried different experimental parameters (other than the reported procedure) to optimize GDY?

Response: In the synthesis process, coppers foils were used as the catalyst.

In addition to the reported procedure, we have also tried many different experimental parameters (e.g., different synthesis temperatures, HEB concentrations, reaction times, and solvent types, etc.) to optimize GDY.

2. Line 95, the authors report a very narrow size distribution (with a distribution of 0.03 nm). This does not agree with the histogram in Fig. 1a. Please clarify.

Response: The size distribution of Rh nanocrystals was calculated based on the Gaussian distribution. The average particle size of Rh nanocrystals was 5.19 nm, standard error is 0.03 nm.

3. Lin 140, the authors claim that "there are no changes in the compositions and chemical states". However, the surface peaks of sample 1 are wider and show higher binding energy. Please discuss the reason.

Response: Thanks for your concern.

For XPS measurements, the shape of the XPS peaks could be influenced by either the rough surface structure of the catalysts or the surface charging effects during the depth-profiling XPS tests.

In this work, the Rh/GDY catalysts were synthesized through in-situ growth of the Rhodium (Rh) nanocrystals on porous graphdiyne (GDY) surface, resulting in a porous and rough surface morphology. Besides, the Rh nanocrystals show better conductivity than GDY, which might lead to the surface charging effect during the depth-profiling XPS tests.

Corresponding discussion has been provided in the revised manuscript.

4. Line 153, "due to the reducibility of GDY...", "the Rh ions can be easily reduced to metallic species". But then why GDY can oxidize Rh up to Rh³⁺?

Response: Thanks for your concern.

This has been discussed in the "Electrocatalysts synthesis and characterization" section of the manuscript. Briefly, in our experiments, GDY was used as the support for the in-situ growth of the Rh nanocrystals, including the adsorption of Rh ion on the surface of GDY and the following in-situ nucleation and growth of Rh nanocrystals. Due to the reducibility of GDY itself and HCOOH, the Rh ions can be easily reduced to metallic species. Besides, our experimental results demonstrated that the presence of strong charge transfer ability between Rh and GDY, which further induce the formation of Rh species with higher valence states.

5. Line 157, what are the "typical semiconductor properties"? Does it indicate a bandgap and low electrical conductivity of the mixed-valence states?

Response: The catalyst obtained in this work has the mixed-valence states. This catalyst has the semiconductor property.

This has been updated in the revised manuscript.

6. The calculation shows that the Volmer–Tafel process is likely the reaction path. However, a ~ 0.20 eV energy barrier exhibits in the Tafel process. With such a barrier, please discuss how can the Rh/GDY drive the 1 A/cm^2 current density at 65 mV?

Response: The HER on stepped Rh/GDY proceeds most likely *via* the Volmer–Tafel mechanism, which is according to the experimental results with a low Tafel slope of 21 mV dec^{-1} . The calculations further confirm that the Volmer–Tafel process is the most preferred among various possible pathways towards HER on the stepped Rh/GDY according to the free energy diagrams. In addition, Gibbs free energy for hydrogen adsorption (ΔG_{H^*}) is a generally acknowledged descriptor to determine the HER activity of a catalyst. A highly active HER sites of catalyst must comply with the thermal-neutral requirement ($\Delta G_{\text{H}^*} \approx 0$), where the atomic hydrogen binding mode is neither too strong nor too weak to allow concomitant efficient hydrogen adsorption and release. The calculated ΔG_{H^*} value for stepped Rh/GDY is only 0.02 eV, which is extremely close to the thermal-neutral state. Thereby, the Rh/GDY catalyst can drive the large current density with a low barrier.

Thank you again for your positive comments on our manuscript.

Yours Sincerely,

Yuliang Li

Professor

Institute of Chemistry, Chinese Academy of China, Beijing, 100190, P. R. China

Tel: +86 10 62587552

Fax: +86 10 82616576

E-mail: ylli@iccas.ac.cn

REVIEWERS' COMMENTS

Reviewer #1 (Remarks to the Author):

The author had answered most of my comments. I recommend it for publication as is.

Reviewer #2 (Remarks to the Author):

I have read the authors' response to mine as well as the other three reviewers' comments carefully. Overall I think the authors have made significant more works and done a nice job in addressing these points. The work can be useful and very instructive to the community with clear application potential in practical sea water splitting. I am therefore very supportive to recommend its publication in Nature Communications after three very minor points below which the authors are suggested to improve.

(1). The authors used "high current density" or "large current density" in the manuscript, please make it consistent.

(2). Regarding iR-compensation in Figure 2b, did the authors used a 85% compensation or larger/smaller? Please write the details in the manuscript so that readers can better understand the plot.

(3). The reference sequence seems to be still not correct, e.g., ref 16 is not about C-CoPx. This may happen when the authors used some softwares for referencing. Please double check this point and correct it.

Reviewer #3 (Remarks to the Author):

The authors have correctly addressed the comments from the reviewers and the manuscript should be ready for publication.

Minor comments are:

- The added paragraph "From the further perspective of mass hydrogen production through water splitting, the electrocatalysts should have the following necessary properties..." contains quite general properties for an efficient HER, but authors could consider to add one or two specific properties needed for efficient and stable HER under saline water conditions.
- Figure 5 caption: H₂O in line 2 should have 2 in subindex.
- Methods: sentence "In the synthesis process, coppers foils were used as the catalyst" should read "copper foils"

This is a very nice piece of work and I would like to congratulate the authors for the thorough revision.

Reviewer #4 (Remarks to the Author):

The authors addressed some of the concerns but the following ones are still not convincing.

1. The size distribution is typically plotted as average +/- standard deviation. It is not clear how the "standard error of 0.03 nm" was calculated. From the histogram, the standard deviation appears larger than 0.03 nm.

2. For the role of GDY to both reduce and oxidize Rh, can the authors cite a reference discussing such a phenomenon?

Response to reviewers

Dear Reviewers,

Thank you for the sincere advice and comments on our manuscript entitled "Large porosity of Rh/GDY for high-performance conversion of saline water to hydrogen fuels" (NCOMMS-22-19836A). We are very grateful for your valuable suggestions and comments on the technical details, formatting, structure of the paper that are very helpful for us to improve our work. Based on your revision suggestions, we have made the revision on our manuscript and the detailed point-by-point responses to all comments are listed as follow.

Reviewer #1 (Remarks to the Author):

The author had answered most of my comments. I recommend it for publication as is.

Response: Thanks for your kind assistance.

Reviewer #2 (Remarks to the Author):

I have read the authors' response to mine as well as the other three reviewers' comments carefully. Overall I think the authors have made significant more works and done a nice job in addressing these points. The work can be useful and very instructive to the community with clear application potential in practical sea water splitting. I am therefore very supportive to recommend its publication in Nature Communications after three very minor points below which the authors are suggested to improve.

(1). The authors used "high current density" or "large current density" in the manuscript, please make it consistent.

Response: Thanks. We have changed the "high current density" to "large current density" in the revised manuscript.

(2). Regarding iR -compensation in Figure 2b, did the authors used a 85% compensation or larger/smaller? Please write the details in the manuscript so that readers can better understand the plot.

Response: All linear sweep voltammetry (LSV) curves were corrected with 100% iR -compensation. The details regarding iR -compensation have been provided in the revised manuscript.

(3). The reference sequence seems to be still not correct, e.g., ref 16 is not about C-CoPx. This may happen when the authors used some softwares for referencing. Please double check this point and correct it.

Response: Thank you for your kind reminding.

We have double checked and corrected the reference sequence in the revised manuscript.

Reviewer #3 (Remarks to the Author):

The authors have correctly addressed the comments from the reviewers and the manuscript should be ready for publication.

Minor comments are:

-The added paragraph "From the further perspective of mass hydrogen production through water splitting, the electrocatalysts should have the following necessary properties..." contains quite general properties for an efficient HER, but authors could consider to add one or two specific properties needed for efficient and stable HER under saline water conditions.

Response: Thank you for your kind suggestion.

The properties needed for efficient and stable HER under saline water conditions has been added. Corresponding description is listed as follow.

From the further perspective of mass hydrogen production through water splitting, the electrocatalysts should have the following necessary properties: (i) high intrinsic catalytic activity that can achieve large current density HER at low overpotentials under saline water conditions; (ii) strong corrosion resistance and electrochemical/mechanical stability at large current densities; (iii) efficient gas/mass diffusion ability capable of separating the formed H₂ bubbles to maintain the catalytic activity for large current density HER; (iv) long-term operational stability under practical conditions (e.g., saline water, strong basic water, or seawater); and (v) low cost.

- Figure 5 caption: H₂O in line 2 should have 2 in subindex.

Response: This has been updated in the revised manuscript as follow.

TS transition state (TSs) with the NEB barriers for H₂O dissociation and H migration, respectively.

- Methods: sentence "In the synthesis process, coppers foils were used as the catalyst" should read

"copper foils"

Response: This has been corrected in the revised manuscript as follow.

In the synthesis process, copper foils were used as the catalyst.

This is a very nice piece of work and I would like to congratulate the authors for the thorough revision.

Response: Thanks for the careful review and comments on the technical details, formatting, structure of the paper that are very helpful for us to improve our work.

Reviewer #4 (Remarks to the Author):

The authors addressed some of the concerns but the following ones are still not convincing.

1. The size distribution is typically plotted as average +/- standard deviation. It is not clear how the "standard error of 0.03 nm" was calculated. From the histogram, the standard deviation appears larger than 0.03 nm.

Response: Thanks for your kind reminding. The size distribution of Rh nanocrystals was calculated to be 5.16 ± 1.20 nm (Figure R1). This has been updated in the revised manuscript.

Figure R1. Size distribution of Rh nanocrystals on the surface of GDY (403 Rh nanocrystals were counted).

2. For the role of GDY to both reduce and oxidize Rh, can the authors cite a reference discussing such a phenomenon?

Response: Thanks for your kind suggestion. References discussing such a phenomenon have been cited in the revised manuscript.

Thank you again for your positive comments on our manuscript.

Yours Sincerely,

Yuliang Li

Professor

Institute of Chemistry, Chinese Academy of China, Beijing, 100190, P. R. China

Tel: +86 10 62587552

Fax: +86 10 82616576

E-mail: yli@iccas.ac.cn